# Gaggle: Visual Analytics for Model Space Navigation

Subhajit Das
Georgia Institute of Technology

Dylan Cashman
Tufts University

Remco Chang
Tufts University

Alex Endert
Georgia Institute of Technology

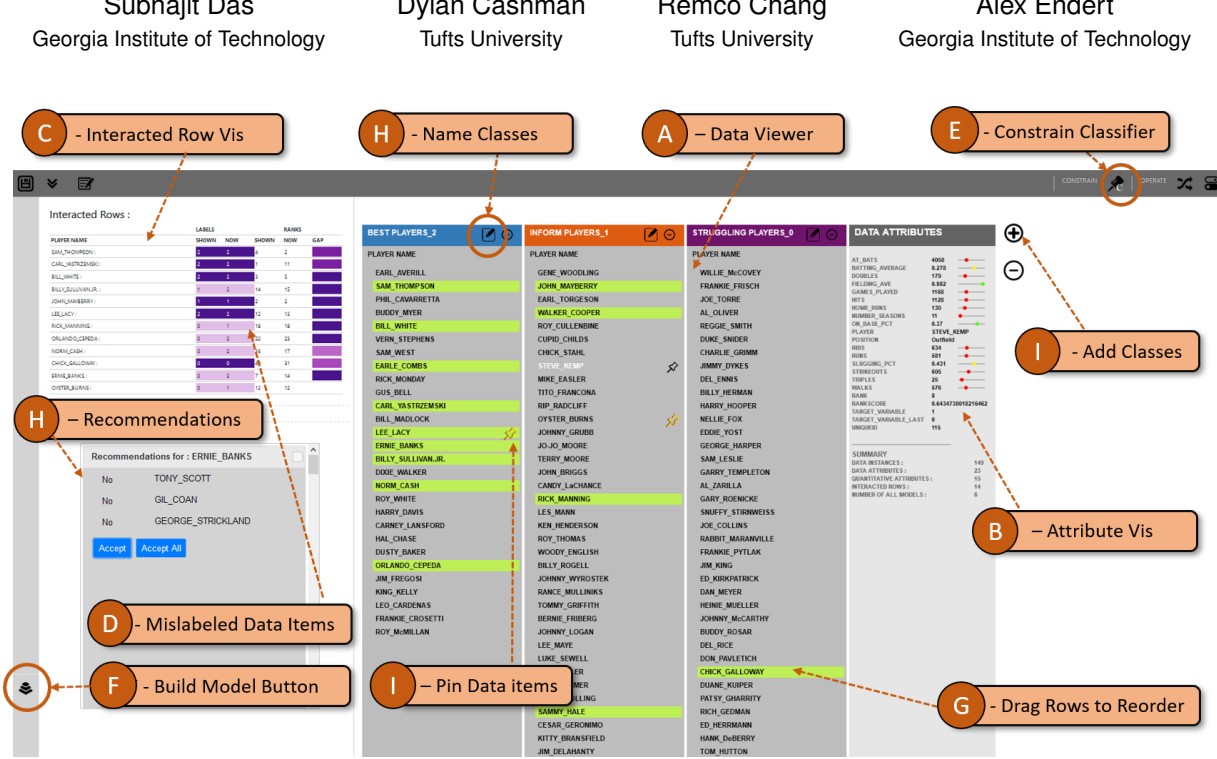

Figure 1: The Gaggle UI allows people to interactively navigate a model space to support interactive classification and ranking of data items. Users can create labels, drag and drop data items into various class labels to specify their subjective preferences to construct classification and ranking models.

## ABSTRACT

Recent visual analytics systems make use of multiple machine learning models to better fit the data as opposed to traditional single, pre-defined model systems. However, while multi-model visual analytic systems can be effective, their added complexity adds usability concerns, as users are required to interact with the parameters of multiple models. Further, the advent of various model algorithms and associated hyperparameters creates an exhaustive model space to sample models from. This poses complexity to navigate this model space to find the right model for the data and the task. In this paper, we present Gaggle, a multi-model visual analytic system that enables users to interactively navigate the model space. Further translating user interactions into inferences, Gaggle simplifies working with multiple models by automatically finding the best model from the high-dimensional model space to support various user tasks. Through a qualitative user study, we show how our approach helps users to find a best model for a classification and ranking task. The study results confirm that Gaggle is intuitive and easy to use, supporting interactive model space navigation and automated model selection without requiring any technical expertise from users.

**Index Terms:** Human-centered computing—Visualization—Classification and ranking model visualization—Mixed initiative systems;

## 1 INTRODUCTION

Visual analytic (VA) techniques continue to leverage machine learning (ML) to provide people effective systems for gaining insights into data [28]. Systems such as Interaxis [39] help domain experts combine their knowledge and reasoning skills about a dataset or domain with the computational prowess of machine learning. These systems are traditionally designed with a pre-defined single ML model that has a carefully chosen learning algorithm and hyperparameter setting. Various combination of learning algorithms and hyperparameters give rise to a vast number of different model types (see Table 1). These different models constitute an exhaustive model space from which unique models can be sampled using a distinct combination of a learning algorithm and associated hyperparameters. For example, support vector machine (SVM) models have many options for kernel functions (i.e., linear, poly or radial) and hyperparameters (i.e., $C$ (regularization parameter), $\gamma$ (kernel coefficient), etc.).

When a model is correctly chosen for the phenomena, task, data distribution, or question users try to answer, existing VA techniques can effectively support users in exploration and analysis. However, in cases where the right model (or optimal model, as desired by the user) to use for a problem is not known a priori, one needs to navigate this model space to find a fitting model for the task or the problem. To combat this, recent VA systems use multiple ML models to support a diverse set of user tasks (e.g., Regression, Clustering, etc. [15, 17, 22, 69]). For example, the VA system Clustervision [42] allows users to inspect multiple clustering models and select one based on quality and preference. Similarly, Snowcat [16] allows inspecting multiple ML models across a diverse set of tasks, such as classification, regression, time-series forecasting, etc. However,

these multi-model systems are often more complex to use compared to single-model alternatives (e.g, in Clustervision users need to be well-versed with cluster model metrics and shown models.) We refer to this complex combination of parameter and hyperparameter settings as *model space*, as there are a large number of models that can be instantiated in this hyperdimensional space. Further, the interactive exploration of different parameter and hyperparameter combinations can be referred to as *model space navigation*. Our definition of *model space* is related to the work by Brown et al. [14] where they presented a tool called *ModelSpace* to analyze how the model parameters have changed over time during data exploration.

In this paper we present Gaggle, a visual analytic tool that provides the user experience of a single-model system, yet leverages multiple models to support data exploration. Gaggle constructs multiple classification and ranking models, and then using a bayesian optimization based hyperparameter selection technique, automatically finds a classification and ranking model for users to inspect, thus simplifying the search for an optimal model as preferred by the user. Furthermore, our technique utilises simple user interactions for model space navigation to find the right model for the task. For example, users can drag data items into specific classes to record classification task's user preferences. Similarly, users can demonstrate that specific data items should be higher or lower in rank within a class by dragging them on top of each other.

Gaggle uses ML to help users in data exploration or data structuring tasks, e.g, grouping data in self-defined categories, and ranking the members of the group based on their representativeness to the category. For example, a professor may want to use a tool to help categorize new student applications in different sets, and then rank the students in each set. Similarly, a salesman may want to cluster and rank potential clients in various groups. These problems fall under classification tasks in ML; however, unlike a conventional classification problem, our use case specifically supports interactive data exploration or data structuring, the models constructed are not meant to predict labels for unseen data items in future. Using this workflow, we expect our technique guards against possible model overfitting incurred due to adjusting the models to confirm to specified user preferences. Furthermore, Gaggle addresses a common problem of datasets that either lack adequate ground truth, or do not have it [54, 67, 80]. To resolve this problem, Gaggle allows users to iteratively define classes and add labels. On each iteration, users add labels to data items and then build a classifier.

We conducted a qualitative user study of Gaggle to collect user feedback on the system design and usability. The results of our study indicate that users found the workflow in Gaggle intuitive, and they were able to perform classification and ranking tasks effectively. Further, users confirmed that Gaggle incorporated their feedback into the interactive model space navigation technique to find the right model for the task. Overall, the contributions of this paper include:

- A model space navigation technique facilitated by a Bayesian optimization hyperparameter tuning and automated model selection approach.
- A VA tool Gaggle, that allows interactive model space navigation supporting classification and ranking tasks using simple demonstration-based user interactions.
- The results of a user study testing Gaggle's effectiveness to interactively build classifiers and ranking models.

## 2 RELATED WORK

### 2.1 Interactions in Visual Analytics

Interactive model construction is a flourishing avenue of research. In general, the design of such systems makes use of both explicit user interactions such as specifying parameters via graphical widgets (e.g., sliders), or implicit feedback including demonstration-based interactions or eye movements to provide guidance on model selection and steering. These types of systems build many kinds of models, includ-

ing classification [7, 32], interactive labeling [18], metric learning [15], decision trees [72], and dimensional reduction [27, 39, 43]. For example, Jeong et al. presented iPCA to show how directly manipulating the weights of attributes via control panels helps people adjust principal component analysis [36]. Similarly, Amershi et al. presented an overview of interactive model building [4]. Our work differs from these works in two primary ways. First, our technique searches through multiple types of models (i.e., Random Forest models with various hyperparameter settings for classification and ranking tasks). Second, our tool interprets user interaction as feedback on the full hyperparameter space using bayesian optimization, causing hyperparameter tuning directly changing model behavior *in parallel*. Stumpf et al. conducted experiments to understand the interaction between users and machine learning based systems [65]. They found that user feedback included suggestions for re-weighting of features, proposing new features, relational features, and changes to the learning algorithm. They showed that user feedback has the potential to improve ML systems, but that learning algorithms need to be extended to assimilate this feedback [64].

Interactive model steering can also be done via demonstration-based interaction. The core principle in these approaches is that users do not adjust the values of model parameters directly, but instead visually demonstrate partial results from which the models learn the parameters [13, 15, 25–27, 31, 44]. For instance, Brown et al. showed how repositioning points in a scatterplot could be used to demonstrate an appropriate distance function [15]. It saves the user the hassle to manipulating model hyperparameters directly to reach their goal. Similarly, Kim et al. presented InterAxis [39], which showed how users could drag data objects to the high and low locations on both axes of a scatterplot to help them interpret, define, and change axes with respect to a linear dimension reduction technique. Using this simple interaction, the user can define constraints which informed the underlying model to understand how the user is clustering the data. Wenskovitch and North used the concept of observation level interaction in their work by having the user define clusters in the visualized dataset [76]. By visually interacting with data points, users are able to construct a projection and a clustering algorithm that incorporated their preferences. Prior work has shown benefits from directly manipulating visual glyphs to interact with visualizations, as opposed to control panels [11, 26, 38, 46, 56, 59].

Active Learning (AL) appears similar to techniques used in Gaggle, yet has a few distinct differences. AL is often used in supervised learning problems (e.g., classification) where adequate annotations are not available in the training data, thereby the algorithm selectively seeks labels for a set of informative training examples [35, 48, 53, 81]. Standard AL processes assumes that an oracle (usually a user) can provide accurate labels or annotations for any queried data sample [19, 61, 70]. From a UI perspective, the work presented in this paper aligns closely with both AL and demonstration-based techniques. Gaggle's interaction design lets users manipulate the visual results of the models, interactively add labels to the training set to incrementally navigate the model space. However, the data items users label are not selected by the system, but by users during exploration.

### 2.2 Multi-Model Visual Analytic Systems

Current visual analytics systems focus on allowing the user to steer and interact with a single model type. However, recent work has explored the capability for a user to concurrently interact with multiple models. These systems implement a multi-model steering technique which facilitates the adjustment of model hyperparameters to incrementally construct models that are better suited to user goals. For instance, Das et al. showed interactive multi-model inspection and steering of multiple regression models [22]. Hypertuner [69] looked at tuning multiple machine learning models' hyperparameters. Xu et al. enabled user interactions with many models,

Table 1: User tasks, learning algorithms, hyperparameters, and parameters in Gaggle.

| Tasks | Learning Algorithm | Hyper-parameters | Parameters |
|---|---|---|---|
| **Classif-ication** | Random Forest | Criteria
Max Depth
Min Samples | Attribute Entropy, Information Gain |
| **Ranking** | Ranking Random Forest | Criteria
Max Depth
Min Samples | Attribute Entropy, Information Gain |

but instead of each model, users interacted with ensemble models through multiple coordinated contextual views [77]. Dingen et al. built RegressionExplorer that allowed users to select subgroups and attributes (rows and columns) to build regression models. However, their technique does not weight the rows and columns; they only select 0 or 1 [23]. Mühlbacher et al. showed a technique to rank variables and pairs of variables to support multiple regression model's trade-off analysis, model validation, and comparison [47]. HyperMoVal [32] addressed model validation of multiple regression models by visualizing model outputs through multiple 2D and 3D sub-projections of the n-dimensional function space [52].

Kwon et al. [42] demonstrated a technique to visually identify and select an appropriate cluster model from multiple clustering algorithms and parameter combinations. Clusterophile 2 [17] enabled users to explore different choices of clustering parameters and reason about clustering instances in relation to data dimensions. Similarly, StarSpire from Bradel et al. [13] showed how semantic interactions [26] can steer multiple text analytic models. While effective, their system is scoped to text analytics and handling text corpora at multiple levels of scale. Further, many of these systems target data scientists, while Gaggle is designed for users who are non-experts in ML. In addition, our work focuses on tabular data. It supports interactive navigation of a model space within two classes of models (classification and ranking) by tuning hyperparameters of each of these types of models.

### 2.3 Human-Centered Machine Learning

Human-Centered Machine Learning focuses on how to include people in ML processes [4–6, 58]. A related area of study is the modification of algorithms to account for human intent. Sacha et al. showed how visual analytic based processes can allow interaction between automated algorithms and visualizations for effective data analysis [58]. They examined criteria for model evaluation on an interactive supervised learning system. The found users evaluate models by conventional metrics, such as accuracy and cost, as well as novel criteria such as unexpectedness. Sun et al. developed Label-and-Learn, allowing users to interactively label data [66]. Their goal was to allow users to determine a classifier's success and analyze the performance benefits of adding expert labels [66]. Many researchers have emphasized the knowledge generation process of users performing labeling tasks [9, 10, 24]. Ren et al. explained debugging multiple classifiers using an interactive tool called Squares [55].

Holzinger et al. discussed how automatic machine learning methods are useful in numerous domains [33]. They note that these systems generally benefit from large static training sets, which ignore frequent use cases where extensive data generation would be prohibitively expensive or unfeasible. In the cases of smaller datasets or rare events, automatic machine learning suffers from insufficient training samples, which they claim can be successfully solved by interactive machine learning leveraging user input [33, 34]. Crouser et al. further formalize this concept of computational models fostering human and machine collaboration [20].

### 2.4 Model Space Navigation

We looked at notable works from the literature which supports model space navigation or visualization to understand the current state better. Sedlmair et al. [60] defined a method of variation of model parameters, generating a diverse range of model outputs for each such combination of parameters. This technique called visual parameter analysis investigated the relationship between the input and the output within the described parameter space. Similarly, Pajer et al. [49] showed a visualization technique for visual exploration of a weight space which ranks plausible solutions in the domain of multi-criteria decision making. However, this technique does not explicitly allow navigating models by adjusting hyperparameters but instead varies weightings of user-defined criteria. Boukhelifa et al. explored model simulations by reducing the model space, then presenting it in a SPLOM and linked views [12]. While Gaggle demonstrates an implicit parameter space exploration, this implements an explicit parameter space.

### 2.5 Automated Model Selection

Model building requires selecting a model type, finding a suitable library, and then searching through the hyperparameter spaces for an optimal setting to fit their data. For non-experts, this task can amount to many iterations of trial and error. In order to combat this guessing game, non-experts could use automated model selection tools such as AutoWeka [41, 68], SigOpt [50], HyperOpt [8, 40], Google Cloud AutoML [45], and AUTO-SKLEARN [30]. These tools execute intelligent searches over the model space and hyperparameter spaces, providing an optimal model for the given problem type and dataset. However, these tools are all based on optimization of an objective function which takes into account only features or attributes that are quantifiable, often ignoring user feedback. Instead, our work explores how to incorporate domain expertise into an automated model selection process supported by interactive navigation of the model space.

### 3 USAGE SCENARIO

Gaggle allows users to assign data points to classes and then partially order data items within the classes to demonstrate classification and ranking. Next, the system responds by constructing a model space, then samples multiple variants of classification and ranking models from it. Gaggle searches various sub-regions of the model space to automatically find an optimal classification and ranking model based on model performance metrics (explained later in the paper). Users can iterate using Gaggle by triggering it to construct new models. In this process users provide feedback to the system through various forms of interaction (e.g., dragging rows, assigning new examples to the class labels, correcting previous labels, etc.). This process continues until the user is satisfied with the model, meaning that the automatically selected model has correctly learned the user's subjective knowledge and interpretation of the data (Figure 2). We present a usage scenario to demonstrate the type of problem being solved and the general workflow of the tool.

**Problem Space:** Imagine Jonathan runs a sports camp for baseball players. He has years of experience in assessing the potential of players. He not only understands which data features are important but also has prior subjective knowledge about the players. Jonathan wants to categorize, and rank the players into various categories ("Best Players", "In-form Players", and "Struggling Players") based on their merit.

**User-Provided Labeling:** Jonathan starts by importing the dataset of baseball players in Gaggle, data publicly available from OpenML [73]. The data contains 400 players (represented as rows) and 17 attributes of both categorical and quantitative types. The dataset does not have any ground truth labels. He sees the list of all the players in the Data Viewer (Figure 3-B). He creates the three classes mentioned above and drags respective players in these bins

or classes to add labels. Knowing *Carl Yastrzemski* as a very highly rated player, he places him in the "Best Players" class. Gaggle shows him recommendations of similar players for labeling (Figure 3).

**Automated Model Generation:** Jonathan clicks the build model button from the Side Bar (see Figure 1-F). Based on Jonathan's interaction so far, Gaggle constructs the model space comprising of multiple classification and ranking models. Gaggle runs its optimizer to navigate the model space based on Jonathan's interaction to automatically find the best performing model, out of an exhaustive search of over 200 random forest models. When the system responds, he finds player *Ernie Banks* is misclassified. He places this player in the "In-form Players" class instead of the "Struggling Players". He moves *Ernie Banks* and similar other misclassified players to the correct class and asks Gaggle to find an optimal model that takes his feedback into account.

Gaggle updates its model space based on the feedback provided by Jonathan, and samples a new classification and ranking model. Gaggle updates the Data Viewer with the optimal model's output. Jonathan reviews the results to find that many of the previously misclassified players are correctly labeled and pins them to ensure they do not change labels in future iterations. Next, he looks at the Attribute Viewer (Figure 1-B) in search of players with high "batting average" and "home runs". He moves players that match his criteria into respective labels (e.g., placing *Sam West* and *Bill Madock* in the "In-Form Players" class). After Gaggle responds with a new optimal model, he verifies the results returned by the model in the interacted row visualization (Figure 1-C). He accepts the classification model and moves on to rank the players within each class.

Jonathan specifies examples for the ranking model by dragging players in each class up or down. After showing a set of relative ranking orderings between data instances (green highlights show interacted items), he iterates to check the full data set, as ranked by Gaggle. He moves player *Norm Cash* and *Walker Cooper* to the top of the "struggling players" class, and moves player *Hal Chase* in the "best players" class. Observing that some data items are not ranked as expected, he further specifies other ranking examples, and triggers Gaggle to construct a new ranking model. Finally, Jonathan finds Gaggle ranked most of the players at the correct spot. In this scenario, we showed how Gaggle helps a domain expert navigate the model space to classify and rank data items solely based on his prior subjective domain knowledge, following the iterative process shown in Figure 2.

While this use case presented how Gaggle could be used by domain experts with a specific dataset, there are other datasets which can be utilised with Gaggle to perform classification and ranking. For example, the drug consumption dataset [29], contains personality measurements (e.g., neuroticism, extraversion, openness to experience, agreeableness, and conscientiousness), level of education, age, gender, ethnicity, etc. of 1885 respondents. Using this dataset a narcotics expert/analyst can classify and rank the respondents into one of the seven class categories (in relation to drug use), namely: "Never Used","Used over a Decade Ago", "Used in Last Decade", "Used in Last Year", "Used in Last Month", "Used in Last Week", and "Used in Last Day". They can also create new classes and rank the respondents based on the attribute values and their prior knowledge. Similarly Gaggle can support a loan officer to use the credit card default payment dataset [79] to decide on approval of loan applications by classifying and ranking the applicants.

## 4 GAGGLE: SYSTEM DESCRIPTION

The overarching goal driving the design of Gaggle is to let people interactively navigate a model space of classification, and ranking models in a simple and usable way. More specifically, the design goals of Gaggle are:

**Enable interactive navigation of model space:** Gaggle should allow the exploration and navigation of the hyper-dimensional model space for classifiers and ranking models.

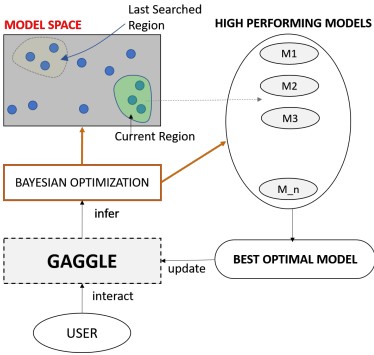

Figure 2: The gray box on the top shows the model space from which candidate models are sampled and ranked based on metrics derived from user interactions, ultimately selecting and showing a single model.

**Support direct manipulation of model outputs:** Model outputs should be shown visually (lists for ranking models, and bins for classifiers). User feedback should directly adjusting data item ranking or class membership, not adjusting model hyperparameters directly.
**Generalize user feedback across model types:** User feedback to navigate the model space should not be isolated on any specific type of model. For instance, providing visual feedback to the classification of data points might also adjust the ranking of data items.
**Leverage user interaction as training data:** User feedback on data points should serve as training data for model creation. Data items interacted with will serve as the training set, and performance is validated against the remaining data for classification and ranking.

### 4.1 User Interface

**Data Viewer:** The main view of Gaggle is the Data Viewer, which shows the data items within each class (Figure 1-A). Users can add, remove, or rename classes at any point during data exploration and drag data instances to bins to assign labels. Users can re-order instances by dragging them higher or lower within a bin to specify relative ranking order of items. Gaggle marks these instance with a green highlight, see Figure 1-G. When Gaggle samples models from the model space and finds an optimal model, the Data Viewer updates the class membership and ranking of items to reflect the models' output. Our design decision to solve for a single model to show at each iteration is to simplify the user interface by removing a model comparison and selection step.
**Attribute Viewer:** Users can hover over data items to see attribute details (Figure 1-B) on the right. Every quantitative attribute is shown as a glyph on a horizontal line. The position of the glyph on the horizontal line shows the value of the attribute in comparison to all the other data instances. The color encodes the instance's attribute quality in comparison to all other instances (i.e., green, yellow, and red encodes high, mid, and low values respectively).
**Data Recommendations:** When users drag data instances to different bins, Gaggle recommends similar data instances (found using a cosine distance metric), which can also be added (Figure 3 and 1-H). This is to expedite the class assignment during the data exploration process. The similarity is computed based on the total distance $D_a$ of each attribute $d_i$ of the moved data instance to other instances in the data. Users can accept or ignore these recommendations.
**Interacted Row Visualization:** This view (Figure 1-C) shows the list of all interacted data items. In addition, with color encoding it shows correct label matches (shown in blue color) and incorrect label matches(shown in pink color). Same is true for ranking (blue for correct ranked order prediction as expected and pink for otherwise). It shows how many constraints were correctly predicted.

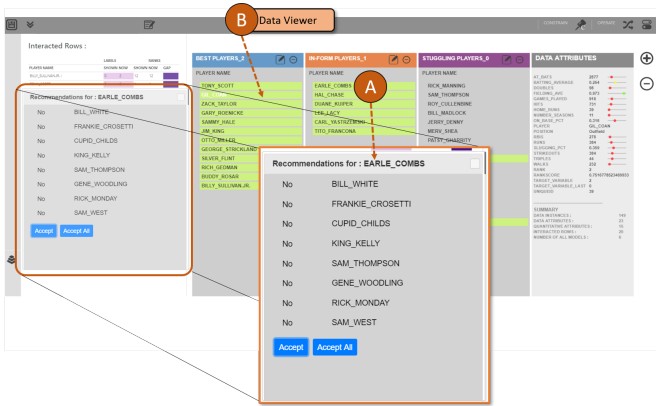

Figure 3: Gaggle's recommendation dialog box.

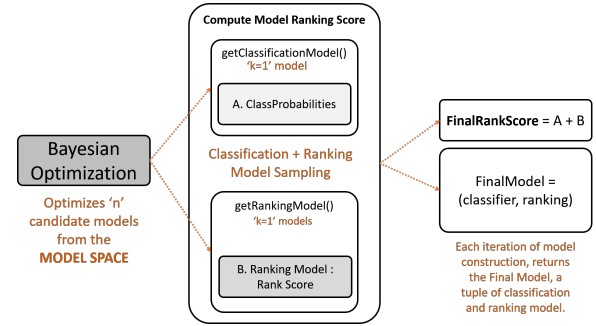

Figure 4: The model ranking method uses Bayesian optimization solver to rank candidate models from the model space.

**User Interactions:** Gaggle lets users give feedback to the system to sample models in the next iteration, adjust model parameters and hyperparameters, and allow users to explore data and gain insight.

- **Assign class llabels:** Users can reassign classes by dragging data items from one class to another. They can also add or remove classes. These interactions provide more constraints to steer the hyperparameters of the classifier.
- **Reorder items within classes:** Users can reorder data items within classes, (see Figure 1-G) to change their ranking. This interaction helps users exemplify their subjective order of data instances within classes. This feedback is incorporated as training data for the ranking model.
- **Pin data items:** When sure of a class assignment of a data item, the user can pin it to the respective class bin (see Figure 1-I). It ensures that data item will always be assigned that class in every subsequent iteration.
- **Constrain classifier:** When satisfied by the classifier, users can constrain the last best classifier. It allows users to move on to show ranking examples for Gaggle to focus on improving the ranking model (Figure 1-C).

## 5 TECHNIQUE

**Models:** We define a *model* as a function $f : \mathcal{X} \mapsto \mathcal{Y}$, mapping from the input space $\mathcal{X}$ to the prediction space $\mathcal{Y}$. We are concerned primarily with *semi-supervised learning* models, in which we are provided with a partially labeled or unlabeled training set $D_{train} = D_U \cup D_L$, where $D_L$ is labeled data and $D_U$ is unlabeled data such that if $d_i \in D_L$, then $d_i = (\mathbf{x_i}, y_i)$, and if $d_i \in D_U$, then $d_i = (\mathbf{x_i})$, where $\mathbf{x_i}$ are features and $y_i$ is a label. A *learning algorithm A* maps a training set $D_{train}$ to a model $f$ by searching through a parameter space. A model is described by its *parameters* $\theta$, while a learning algorithm is described by its *hyperparameters* $\lambda$. A model parameter is internal to a model, where its value can be estimated from the data, while model hyperparameters are external to the model.

**Model Space:** Varying the learning algorithm and the hyperparameters creates a diverse set of new models. This space of every possible combination of learning algorithms and hyperparameters forms a high dimensional *model space*. The objective to find an optimal model from this high-dimensional, infinitely large space without any computational guidance or statistical methods is similar to finding a needle in a haystack. Conventionally, ML practitioners/developers navigate the model space using data science principles to test various candidate models. They search for regions (sub-space of the model space) to find optimal models. For instance, one can navigate the model space by randomly sampling new models and testing their performance in terms of accuracy (or other defined metrics) to find a model that best suits the task.

Gaggle constructs a model space by sampling multiple random

forest models which takes a predefined list of hyperparameters (criteria, max depth, and min samples to set a node as a leaf) within a set domain range (see Table 1). While Gaggle uses a random forest model for the system evaluation, the general optimization method used is designed to work with other learning algorithms and hyperparameter combinations as well. For instance, Gaggle's optimizer can sample multiple SVM models using a set of chosen hyperparameters such as $C$ (regularization parameter), $\gamma$ (kernel coefficient).

### 5.1 Interactive model space navigation

To facilitate interactive user feedback and navigation of the model space, Gaggle uses a Bayesian optimization technique [51, 62]. This navigation is initiated by randomly sampling models from the model space as shown in Figure 5. Gaggle seeds the optimization technique by providing: a learning algorithm $A$, a domain range $D_r$ for each hyperparameter, and the total number of models to sample $n$ for both classification and ranking models. Gaggle uses a Bayesian optimization module that randomly picks a hyperparameter combination $hp_1$, $hp_2$ and $hp_3$. For example, a model $M_1$ can be sampled by providing "learning algorithm" = "random forest", "criteria type" = *gini*, "max-depth" = 30, and "min-samples-leaf" = 12. Likewise, the Bayesian optimization module samples $M_1, M_2, M_3, M_4 \dots M_n$ models. For each model, it also computes a score $S_i$ based on custom-defined model performance metrics inferred from user interactions.

The Bayesian optimization module uses a Gaussian process to find an expected improvement point in the search space (of hyperparameter values) over current observations. For example, a current observation could be mapped to a machine learning model, and its metric for evaluation of the expected probability can be precision score or cross-validation score. Using this technique, the optimization process ensures consistently better models are sampled by finding regions in the model space where better performing models are more likely to be found (see Figure 2). Next, the Bayesian optimization module finds the model with the best score $S_i$ (see Figure 5). Gaggle performs this process for both classification and ranking models driven by user-defined performance metrics.

**Classification Model Technique**: Gaggle follows an unconventional classifier training pipeline. As Gaggle is designed to help users in data exploration using ML, and not to make predictions on unseen data items, the applicability of conventional train and test set does not apply. Gaggle begins with an unlabeled dataset. As the user interacts with an input dataset $D$ of $n$ items, labels are added; e.g., if the user interacts with $e$ data items, they become part of the training set for the classification model. The rest of the instances $n - e$, are used as a test set to assign labels from the trained model. If $e$ is lower than a threshold value $t$, then Gaggle automatically finds $s$ similar data instances to the interacted items and places them in the training set along with the interacted data items ($s$ gets the label from the most similar labeled data item in $e$). The similarity

is measured by the cosine distance measure using the features of the interacted samples. This ensures that there are enough training samples to train the classifier effectively.

As users interact with more data instances, the size of the training set grows, and test set shrinks, helping them to build a more robust classifier. For each classifier, Gaggle determines the class probabilities $P_{ij}$, representing the probability of data item $i$ classified into class $j$. The class probability is used to augment the ranking computation (explained below) as they represent the confidence the model has over a data item to be a member of a said class. Gaggle's interactive labeling approach has close resemblance to active learning (AL) [78], where systems actively suggest data items users should label. Instead, Gaggle allows users to freely label any data item in $D$ to construct a classifier. Furthermore, our technique incorporates user feedback to both classify and rank data items.

**Ranking Model Technique**: Gaggle's approach to aid interactive navigation of the model space for the ranking task is inspired by [37, 74]; which allows users to subjectively rank multi-attribute data instances. However, unlike them, Gaggle constructs the model space using a random forest model (a similar approach to [82]) to classify between pairs of data instances $R_i$ and $R_j$. While we tested both of these approaches, we adhered to random forest models owing to it's better performance with various datasets. Using this technique, a model predicts if $R_i$ should be placed above or below $R_j$. It continues to follow the same strategy between all the interacted data samples and the rest of the data set. Further, Gaggle augments this ranking technique with a feature selection method based on the interacted rows. For example, assume a user moves $R_i$ from rank $B_i$ to $B_j$ where $B_i > B_j$ (the row is given a higher rank) Our feature selection technique checks all the quantitative attributes of $R_i$, and retrieves $m = 3$ (the value of $m$ is learnt by heuristics and can be adjusted) attributes $Q = Q_1, Q_2$, and $Q_3$ which best represents why $R_i$ should be higher in rank than $R_j$. The attribute set $Q$ are the ones in which $R_i$ is better than $R_j$. If $B_i < B_j$ then Gaggle retrieves features that supports it and follows the above protocol.

This technique performs the same operations for all the interacted rows, and finally retrieves a set of features ($F_s$, by taking the common features from each individually interacted row) that defines the user's intended ranked order. In this technique, if a feature satisfies one interaction but fails on another, they are left out. Only the common features across interacted items get selected. If the user specifies incoherent data instances that leads to no or very small set in $F_s$, Gaggle uses SK Learn's K Best feature selection technique to fill $F_s$. However, this may produce models which do not adhere to the shown user interactions.

The set of selected features $F_s$ are then used to build the random forest model for the ranking task which computes a ranking score $E_{ij}$ ($i$th instance, of $j$th class) for each data item in $D$. Next, using the class probabilities $P_{ij}$ and the ranking score $E_{ij}$, Gaggle ranks the data instances within each class. A final ranking score $G_{ij} = E_{ij} * W_r + P_{ij} * (1 - W_r)$ is computed by combining the ranking score $E_{ij}$ of each data item in $D$ and its class probability $P_{ij}$, retrieved from the classifier, where $W_r$ is the weight of the rank score and $1 - W_r$ is the weight of the classification probability (see Figure 4). The weights are set based on the model accuracy on various datasets. Finally the dataset is sorted by $G_{ij}$. While the described technique uses random forest models, in practice we have tested it with other ML models such as SVM. Furthermore, the weights described here are a set of hyperparameters that needs to be tuned based on the chosen model and the dataset.

## 5.2 Model Selection

Gaggle selects an optimal model from the model space based on the following metrics which describe each model's performance (see Figure 4). These are fed to the Bayesian optimization module to sample better models:

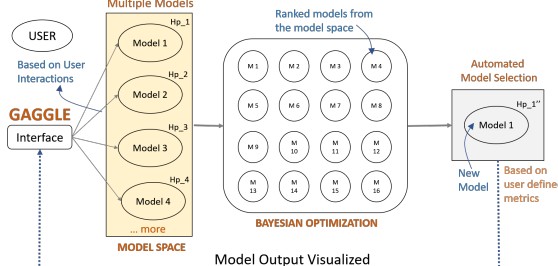

Figure 5: Model space navigation approach using Bayesian optimization to find the best performing model based on user-defined metrics.

**Classification Metrics:** Metrics used to evaluate the classifiers include: percentage of wrongly labeled interacted data instances $C_u$, and cross-validation scores from 10-fold evaluation $C_v$ (both range between $0 - 1$). Other metrics such as precision, F1-score, can be specified based on the dataset and the user's request. The final metric is the sum total of these components computed as: $C_u * W_u + C_v * W_v$ where, $W_u$ and $W_v$ are the respective weights for each of the aforementioned classification metric components. Different weight values were tested during implementation and testing. We chose the set of weights which led to the best gain in model accuracy.

**Ranking Metrics:** To evaluate models for the ranking task, Gaggle computes three ranking metrics based on the absolute distance from a data instance's position before and after a said model $M_i$ is applied to the data. Assume a row $r$ is ranked $q$ when the user interacted with the data. After applying model $M_i$ to the data, the row $r$ is at position $p$, then the absolute distance is given by $d_r = abs(p - q)$. The first ranking metric computes the absolute distances only between the interacted rows. It is defined as $Z_u = (\sum_{r \in I} d_r)/l$, where row $r$ is in the set $I$ of all $l$ interacted rows. The second metric, $D_v$, computes the absolute distance between the interacted rows $I$ and the immediate $h$ rows above and below of each interacted rows. It is defined as $Z_v = (\sum_{r \in I_l} (\sum_{t \in H_h} d_{tr})/h)/l$ where row $r$ is in the set $I$ of all $l$ interacted rows, $H$ is the set of $h$ rows above and below of each interacted row $I$. This metric captures if the ranked data item is placed in the same neighborhood of data items as intended by the user. In Gaggle, $h$ defaults to 3 (but could be adjusted). The third metric, $D_w$, computes the absolute distance between all the instances of the data before and after a model is applied. defined as $Z_w = (\sum_{r \in D_n} d_r)/n$ where row $r$ is in the set $D_n$ of all $n$ rows. A lower distance represents a better model fit.

The final ranking metric is computed by the weighted summation of these metrics defined as $Z_{total} = Z_u * W_u + Z_v * W_v + Z_w * W_w$, where, $W_u$, $W_v$, $W_w$ are the weights for the three ranking metrics. Weights were tested during implementation and chosen based on the set of weights which gave the best model accuracy. While $Z_u$ captures user-defined ranking interactions in the current iteration, $Z_v$ and $Z_w$ both ensure that user's progress (over multiple interations) is preserved in ranking the entire dataset. Furthermore, we used these metrics instead of other ranking metrics such as, *normalized discounted cumulative gain* (NDCG) [75], as the latter relies on document relevance, which in this context seemed less useful to capture user preferences. Also, NDCG is not derived from a ranking function, instead relies on document ranks [71]. Another metric called Bayesian personalized ranking (BPR) by Rendle et al. [57] allows ranking a recommended list of items based on users implicit behavior. However, unlike the use case supported by BPR, our work specifically allows users to rank the data subjectively. Furthermore, unlike BPR, our metric also takes into account negative examples, (i.e., when a data item is ranked lower than the rest).

## 6 USER STUDY

We conducted a user study to evaluate Gaggle's automatic model space navigation technique to support the classification and ranking tasks. Our goal was to get user feedback/responses to Gaggle's system features, design, and workflow. Further, collecting observational data, we wanted to know if our technique helps them to find an optimal model satisfying their goal. We designed a qualitative controlled lab study where participants used Gaggle to perform a set of predefined tasks. In the end, they gave feedback to the system design, usability, and workflow.

### 6.1 Participants

We recruited 22 graduate and undergraduate students (14 male). The inclusion criteria were that participants should be non-experts in ML, and have adequate knowledge of movies and cities (datasets used for the study). None of the participants used Gaggle prior to the study. We compensated the participants with a $10 gift card. The study was conducted in a lab environment using a laptop with a 17-inch display and a mouse. The full experiment lasted 60-70 minutes.

### 6.2 Study Design

Participants were asked to complete 4 tasks: multi-class classification of items (3 classes), ranking the classified data items, binary classification of items, and ranking the classified data items. Participants performed the above 4 tasks on 2 datasets, Movies [3] and Cities [2]. To reduce learning and ordering effects, the order of the datasets and the tasks were randomized. In total, each participant performed 8 tasks, 4 per dataset. We began with a practice session to teach users about Gaggle. During this session, participants performed 4 tasks, which took 15 minutes, and included multi-class classification and ranking, and binary classification and ranking on the Cars dataset [1]. We encouraged participants to ask as many questions as they want to clarify system usability or interaction issues. We proceeded to the experimental sessions only when participants were confident with using Gaggle.

Participants were asked to build a multi-class classifier first. This was followed by a binary classification and ranking task on the same dataset. Then they repeat the same set of tasks on the other dataset. The movies data had 5000 items, with 11 attributes, while the cities dataset had 140 items with 45 attributes. We asked participants to create specific classes for each dataset. For the Movies dataset multi-class labels were *sci-fi*, *horror/thriller*, and *misc*, and *fun-cities*, *work-cities*, and *misc* for the Cities dataset. For the binary classification task, the given labels were *popular* and *unpopular* (Movies dataset), and *western* and *non-western* (Cities dataset).

### 6.3 Data Collection and Analysis

We collected subjective feedback and observational data through the study. We encouraged participants to think aloud while they interacted with Gaggle. During the experiment sessions, we observed the participants silently in an unobtrusive way to not interrupt their flow mitigating *Hawthorne* and *Rosenthal* effects. We audio and video recorded every participant's screen. We collected qualitative feedback through a semi-structured interview comprising of open-ended questions at the end of the study. We asked questions such as *What were you thinking while using Gaggle to classify data items?*, *What was your experience working with Gaggle?*, etc. Furthermore, after each trial per dataset, we asked participants to complete a questionnaire containing likert scale questions. For example, we asked: (1) On a scale of 1 to 5, how successfully the system was learning based on interactions provided? (1 is randomly, 5 is very consistently), (2) On a scale of 1 to 5, how satisfied are you with the classification model output? (1 is not satisfied, 5 is very satisfied), (3) On a scale of 1 to 5, how satisfied are you with the ranking model output? (1 is not satisfied, 5 is very satisfied). Here satisfaction means, how well the underlying model adhered to

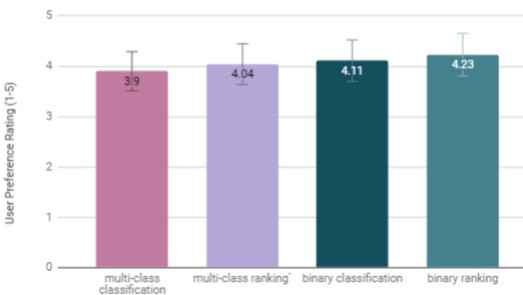

Figure 6: User preferences (averaged over datasets) for the four tasks.

the users demonstrated interactions. Please refer the supplemental material ([1]) to know about the study questionnaire.

### 6.4 User Preferences

We collected user preference rating for all four tasks (see Figure 6). The scores were between $1-5$ (1 meaning least preferred, 5 meaning highly preferred). The average rating of Gaggle for multi-class classification with the ranking task was 3.97. The average rating of Gaggle for the binary classification with the ranking task was 4.17. Though users approved Gaggle's simplicity to allow them to classify and rank data samples, they seemed to prefer Gaggle for the binary classification and ranking task owing to higher accuracy and consistently matching users interpretation of the data.

### 6.5 Model Switching Behavior

For all participants, we collect log data to track how models were selected when users interactively navigated the model space. We sought to understand how model hyperparameters switch during usage. For participants using the Movies dataset (multi-class classification task) the *max-depth* hyperparameter changed values (ranging from 3 to 18). Similarly, for the Cities dataset (multi-class classification task) the hyperparameter *Criteria* ranged from *entropy* to *gini*. The *min-samples* hyperparameter varied within the range of 5 to 36 for both datasets. For the binary classification task, *max-depth* ranged from 4 to 9 for both datasets. Also we noticed the *criteria* hyperparameter switching from *gini* to *entropy* for both datasets for the binary classification task. On average the hyperparameters switched $M = 9.34\ [7.49, 11.19]$ times to support the multi-class classification and ranking task, while the average change was $M = 5.41\ [4.89, 5.93]$ for binary classification and ranking task. These results indicate that the interactive navigation of the model space technique found new models as participants interacted with Gaggle.

### 6.6 Qualitative Feedback

**Drag and drop interaction:** All the participants liked the drag and drop interaction to demonstrate examples to the system. *"I like the drag items feature, it feels very natural to move data items around showing the system quickly what I want."* (P8). However, with a long list of items in one class, it can become difficult to move single items. P18 suggested, *"I would prefer to drag-drop a bunch of data items in a group."*. In future, we will consider adding this functionality.

**Ease of system use:** Most participants found the system easy to use. P12 said *"The process is very fluid and interactive. It is simple and easy to learn quickly."* P12 added *"While the topic of classification and ranking models is new to me, I find the workflow and the interaction technique very easy to follow. I can relate to the use case and see how it [Gaggle] can help me explore data in various scenarios."*

**Recommended items:** Recommending data while dragging items into various labels helped users find correct data items to label. P12 said *"I liked the recommendation feature, which most of the time was accurate to my expectation. However, I would expect something*

---

[1] Data: `https://gtvalab.github.io/projects/gaggle.html`

8

*like that for ranking also."* P2 added *"I found many examples from the recommendation panel. I felt it was intelligent to adapt to my already shown examples."*

**User-defined Constraints:** The interacted row visualization helped users understand the constraints they placed on the classification and ranking models. P14 said *"This view shows me clearly what constraints are met and what did not. I can keep track of the number of blue encodings to know how many are correctly predicted"*. Even though the green highlights in the Data Viewer also mark the interacted data items, the Interacted Row View shows a list of all correct/incorrect matches in terms of classification and ranking.

**Labeling Strategy** Few participants changed their strategy to label items as they interacted with Gaggle. They expected it might confuse the system. However, to their surprise, Gaggle adapted to the interactions and still satisfied most of the user-defined class definitions. P17 said *"In the movies data set, I was classifying sci-fi, and thriller movies differently at first, but later I changed based on recent movies that I saw. I was surprised to see Gaggle still got almost all the expected labels right for non-interacted movies."*

## 7 DISCUSSION AND LIMITATIONS

**Large Model Search Space:** Searching models by combining different learning algorithms and hyperparameters leads to an extremely large search space. As a result, a small set of constraints on the search process would not sufficiently reduce the space, leading to a large number of sub-constrained and ill-defined solutions. Thus, how many interactions are considered optimal for a given model space? In this work, we approached this challenge by using Bayesian optimization for ranking models. However, larger search spaces may pose scalability issues while too many user constraints may "over-constrain" models leading to poor results.

**Scalability:** The current interaction design is intended to support small to moderate dataset sizes. In the user study, we limited the dataset size to understand how users interact with the system and provide feedback to classification and ranking models. However, the current design is not meant to handle cases when the data set is large-ish, i.e., say twenty thousand data items. In the future, we would like to address this concern by using Auto-ML based cloud services coupled with progressive visual analytics [63].

**Abrupt Model and Result Changes:** As users interact to navigate the model space, each iteration of the process may find substantially different models. For example, users might find a random forest model with "criteria = gini" with depth of tree = 50" in one iteration and "criteria = entropy with depth of tree = 2" in the next iteration. This may entail significant changes in the results of these models. While these abrupt changes may not impact users greatly if they are unaware of the model parameterizations, showing users what changed in the output may ease these transition states.

**Data Exploration using ML:** Supporting data exploration while creating models interactively is challenging. Users may change their task definition slightly or learn new information about their data. In these cases, user feedback may be better modeled by a different model hyperparameterization compared to earlier in their task. Updating the class definition or showing better examples impacts the underlying decision boundary, which the classifier needs to map correctly. For example, in earlier iterations, a linear decision boundary might characterize the data. However, when new examples for classes are provided the decision boundary might be better approximated using a polynomial or radial surface (see Figure 7). In situations like this, Gaggle helps users by finding an optimal model with new hyperparameter settings, without changing them manually.

**ML practices and model overfitting:** Conventionally classifiers are trained on a training set, and then validated on a test set. Our technique utilises the full dataset as input data, interacted data items as training set, and the rest as test set. As users iteratively construct classifiers, the training set grows in size and test set reduces. We used

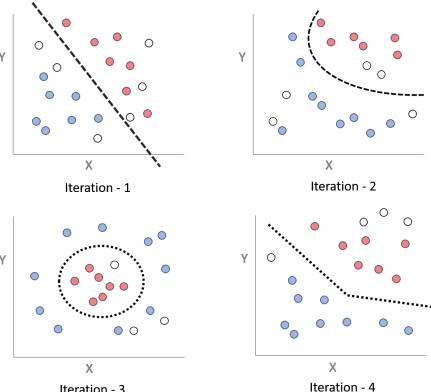

Figure 7: As users gain more knowledge through exploration, they may change their task, which might require different model hyperparameters. (Blue and orange points represent positive and negative classes; white points represent data items not interacted with.)

this approach to account for user-specified preferences through iterative interactions. Nevertheless, our process follows the conventional ML principle, where the classifier training is done independently of the test data. It only makes prediction on it after training, and Gaggle enables users to inspect the results. However, a challenge systems like Gaggle faces is model overfitting [21]. An overly aggressive search through the model space might lead to a model which best serves the user's added constraints, but might underperform on an unseen dataset. We believe that in use cases where ML is utilised to organize or explore the data, the problem of overfitting is less problematic, considering the constructed models are not meant to be used for unseen data.

**Active Learning and Gaggle:** Gaggle's approach to interactive labeling is closely related to active learning (AL) strategies in ML, in which systems request users to specify labels to data instances, on which the model is less confident. However, Gaggle allows freedom in terms of which items to label. AL on the other hand relies on existing labels in the training data and only asks users to re-confirm labels to certain data instances when needed (e.g, when the classifier is less confident on the prediction of a data instance). While the approach incorporated in Gaggle gives users more agency over the process, this approach may be less suitable for larger datasets where AL techniques could present items to users which need feedback.

**Extending the Model Space navigation:** The interactive model space navigation technique that translates user interactions into classification and ranking metrics can be extended to other ML models. For example, other than a random forest model, we have tested Gaggle with SVM model for the classification task and using the RankSVM technique for the ranking task. Likewise, Gaggle can be used with a boosting model for the classification task and a weighted ranking model from each component model from the boosted model for the ranking task.

## 8 CONCLUSION

In this paper, we present an interactive model space navigation approach for helping people perform classification and ranking tasks. Current VA techniques rely on a pre-selected model for a designated task or problem. However, these systems may fail if the selected model does not suit the task or the user's goals. As a solution, our technique helps users find a model suited to their goals by interactively navigating the high-dimensional model space. Using this approach, we prototyped Gaggle, a VA system to facilitate classification and ranking of data items. Further, with a qualitative user study, we collected and analyzed user feedback to understand the usability and effectiveness of Gaggle. The study results show that users agree that Gaggle is easy to use, intuitive,

and helps them interactively navigate the model space to find an optimal classification and ranking model.

## 9 ACKNOWLEDGEMENTS

Support for the research is partially provided by DARPA FA8750-17-2-0107. The views and conclusions contained in this document are those of the authors and should not be interpreted as representing the official policies, either expressed or implied, of the U.S. Government.

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
