# OpenReview forum: "Gaggle: Visual Analytics for Model Space Navigation"
_graphicsinterface.org/Graphics_Interface/2020/Conference — GI 2020_

### Official Review · AnonReviewer1 · 2020-01-02

**Confidence:** 3
**Rating:** 6

**Review:**

In this submission the authors describe Gaggle, a system that takes input from user to facilitate model space navigation in VA contexts. The paper is overall well written (a couple of typos here and there, including some I will report below) and the topic is relevant to GI and the visualization community.

While I was not an expert at all in this domain, I found the paper relatively easy to follow and understand. Not being an expert, I cannot judge whether or not all appropriate previous approaches are cited, but I trust that other reviewers would be able to point out missing references if there are any.

I would overall argue that the work should be accepted provided that the authors can address my (relatively small) concerns (and the ones from other reviewers). I will list my concerns and questions below.

Given the scalability issue that the authors currently highlight, it seems that such a complex system might be an overkill for small datasets. Especially in the way they write about this limitation. I would argue that the authors should somehow justify that their system can be useful in real scenarios despite that limitation and give concrete examples to avoid leaving the reader with this feeling. This is currently my main concern about the submission and the reason why I put a rating a bit lower than 7.

It would be nice to have access to the full set of questions that were asked during the semi-structured interviews, as well as the likert scale questions that were given to participants after each trial. Currently the likert-scale results do not make much sense without being able to see what questions were asked. I overall found the qualitative evaluation to be not correctly reported.

Linked to this, I would argue that the datasets used by the authors do not seem very interesting or complicated. I, so far, fail to see why users would need to use Gaggle to make use of this data. I don’t know if these datasets are representative of the datasets that the authors envision for Gaggle, and I surely hope that they are not, but in this case I would argue that the authors should properly justify why they chose these specific datasets. This is currently missing and it hinders the work that the authors have conducted and reported on previously.


Typos:
Page 3, second column “The found” → “They found”
Page 8 “In future” → “In future work”

---

### Official Review · AnonReviewer3 · 2020-01-06

**Confidence:** 4
**Rating:** 5

**Review:**

This paper presents Gaggle, a visual analytics system that helps novice analysts navigate model space in performing classification and ranking tasks. The system has many features and is probably useful and effective. But there is not much contribution in terms of the visual analytics research or understanding how humans use these types of systems.

There is no doubt in my mind that a lot of work and thoughts have gone into the development of this system. However, mixed initiative systems have been studied for quite a long time. There seems not sufficient novelty in terms of the technical contribution or visualization design in this paper.

First, it is unclear about the effectiveness of the proposed Bayesian based model searching technique. Auto ML has been a hot topic in the machine learning community, e.g., https://sites.google.com/site/automlwsicml14/. This paper does not compare their approach with any other existing methods. It is not convincing that there exists sufficient novelty or contribution. It is also unclear if the proposed method works for navigating any ML model space (e.g., SVM, neutral networks) or just Random Forests (as described in the paper). If this is a limitation of the method, it needs to be discussed.

Moreover, it is not clear who are the end users of Gaggle and whether Gaggle is useful in real world. The presentation of usage scenario is nice. However, it does not come from a real-world use case and I can hardly imagine how Gaggle would contribute to analytical process. It would be fine if the authors collect requirements from target users. The design goals seem to be distilled without involving end users in the loop. This would be okay if there was an insightful section on how human users would interact with such a system based on real user interviews. But the evaluation just uses standard techniques to confirm the usability of this system.

---

### Official Review · AnonReviewer2 · 2020-01-09

**Confidence:** 3
**Rating:** 6

**Review:**

The authors present a new visual analytic system called Gaggle, which aims to enable non-expert users to interactively navigate a model space by using a demonstration-based approach.  An evaluation with 22 non-experts support the claim to simplify the complex model and hyperparameter search by using such an interaction paradigm.

The system is well motivated, its structure is sufficiently described and the overall paper is well written.
However, some open questions and comments remain:

1) The usage scenario is helpful to better understand the application of Gaggle.
However, the difference between the scenario and the example data presented in Figure 1 makes it unnecessary complicated to understand the described scenario in context. I recommend the authors to align these two to improve readability.

2) The last paragraph of the usage scenario is not clear. I recommend to reformulate and clarify this part of the paper.

3) The authors claim that the presented “technique guards against possible model overfitting incurred due to adjusting the models confirm to specified user preferences." (p.2) However, the authors declare later that the risk of overfitting is high with such aggressive model space search approaches like used in Gaggle. While the authors argue further that “ overfitting is less problematic” in an exploratory context, it would strengthen the contribution to discuss potential solutions to this common issue.

4) The author describe the aim of Gaggle as to help users to explore data and gain insights.
However, the process described rather helps users to faster or more accurate build a model that produce intended outcome, similar to active learning approaches.
The contribution would benefit from a reflection and discussion on the active training vs. exploration trade-off.
This could take the form of a more detailed related work analysis regarding active learning and similar approaches and also as part of a larger discussion about the benefits of the presented approach over them.

5) Regarding the presented model, two main question occur: 1) How does the model acts if the users selection is not coherent?
In the paper it is described that “if a feature satisfies one interaction but fails on another, they are left out. Only the common features across interacted items get selected. The set of selected features Fs are then used to build the random forest model”. While no or a very small set of common features might represent an edge case, it is still important to evaluate the robustness and generalizability of the model.
2) The weight selection is described as “The weights are set based on the model accuracy on various datasets.” It would help the reader if the author could elaborate on this aspect and would make the presented approach more replicable by the research community.

6) I encourage the authors to an elaborated discussion of the potential generalizability to other models, contexts and real world scenarios. The current study takes a rather small dataset and exclusively random forest algorithms as an example case, which is quite limited in its application.
To open the contribution of this work to a larger audience, a discussion should include details about necessary changes, limitations of applicability and Gaggle's potential over known approaches.

7) The qualitative evaluation should be described in more detail. This would include which likert scale questions were asked, their results and what did other participants report to present a more comprehensive picture of the overall results (currently only 6/22 referenced).


I encourage the authors to consider the above mentioned comments to improve their submission, especially regarding the difference to other active learning approaches as well as the generalizability to other models and scenarios.
In conclusion, the authors present an interesting approach to help non-experts in ML to consider a diverse set of model parameters, without the burden of setting them manually. The system is well designed for the use case and the study reflects its applicability in this case.

Therefore, I recommend to rather accept this paper, under the condition that the before mentioned comments are considered and addressed.



Spelling mistakes:
- p.2: domainstration-based
- p.9: might require different different model

---

### Meta-Review · Area_Chair1 · 2020-01-11

**Recommendation:** Accept
**Confidence:** 4

**Metareview:**

This paper receives the scores of 6, 5, 6, which positive borderline. In general, this paper is acceptable, while there are several places to improve, especially demonstrating the usefulness and generalization of the system. This paper also misses some details and have several clarity issues. I suggest the authors take the reviewers' comments and revise the paper accordingly. In summary, addressing these concerns should be doable within the review cycle, and thus I recommend to accept this paper.

---

### Decision · Program_Chairs · 2020-01-11

Accept